# Security Analysis of Safe and Seldonian Reinforcement Learning Algorithms

**A. Pinar Ozisik[1]**       **Philip S. Thomas[1]**
[1]College of Information and Computer Sciences
University of Massachusetts
`{pinar, pthomas}@cs.umass.edu`

## Abstract

We analyze the extent to which existing methods rely on accurate training data for a specific class of reinforcement learning (RL) algorithms, known as Safe and Seldonian RL. We introduce a new measure of security to quantify the susceptibility to perturbations in training data by creating an attacker model that represents a worst-case analysis, and show that a couple of Seldonian RL methods are extremely sensitive to even a few data corruptions. We then introduce a new algorithm that is more robust against data corruptions, and demonstrate its usage in practice on some RL problems, including a grid-world and a diabetes treatment simulation.

## 1   Introduction

*Reinforcement learning* (RL) algorithms have been proposed for many high-risk applications, such as improving type 1 diabetes and sepsis treatments [44; 18]. One type of *safe RL* algorithm [41; 43], subsequently referred to as *Safe and/or Seldonian RL* [44], enables these high-risk applications by providing high-confidence guarantees that the application will not cause undesirable behavior like increasing the frequency of dangerous patient outcomes.

However, existing safe RL algorithms rely on the assumption that training data is free from anomalies such as errors, missing entries, and malicious attacks. In real applications, anomalies are common when training data comes from a pipeline that includes human interactions, natural language processing, device malfunctions, etc. For example, the recent application of RL to sepsis treatment in the *intensive care unit* (ICU) used training data generated from hand-written doctors' notes [18]. In a high-stress ICU environment, missing records and poorly written notes are difficult to automatically parse [1]. Furthermore, Petit et al. [35] demonstrated the importance of using reliable training data for self-driving cars, a potential area for the real application of RL. They executed a series of attacks on the camera and sensors of self-driving cars in a lab environment to demonstrate how the safety of passengers can be compromised.

In this paper, we analyze how robust Seldonian RL algorithms are to perturbations in data. Specifically, we analyze the robustness of a specific component, called the *safety test*. This component makes current Seldonian algorithms safe: the safety test checks whether necessary safety constraints are satisfied with high probability. Using data collected from a baseline policy, it outputs new policies that are highly likely to perform at least as well as the baseline. The safety test first computes estimates of the expected performance of a new policy from training data, using *importance sampling* (IS). It then uses *concentration inequalities* (CI) to bound the expectation of the IS estimates.

First, we propose a new measure, which we call $\alpha$-security, for quantifying how robust the safety test of a Seldonian RL algorithm is to data anomalies. To create this measure, we define an attacker that adds adversarially corrupt data points to training data. Although anomalies in data are often not due to an adversarial attacker, if we create algorithms that are robust to adversarial attacks, they

will also be robust to non-adversarial anomalies in data. Second, we analyze the security of existing safety test mechanisms using $\alpha$-security, and find that even if only one data point is corrupted, the high-confidence safety guarantees provided by several Seldonian RL algorithms can be egregiously violated. Then we propose a new algorithm that is more robust to anomalies in training data, ensuring safety with high probability when an upper bound on the number of adversarially corrupt data points is known. Finally, we present experiments that support our theoretical analysis.

Our work is directly applicable to any scenario that requires computing confidence intervals around IS estimates. More broadly, the community is also interested in our definition of safety [8] and its limitations [11], and IS [4; 16; 26; 29]. Lastly, our $\alpha$-security formalization also pertains to high-confidence methods that do not use IS [41; 21; 22], and can be used as a general framework to study their robustness to data corruption attacks.

## 2  Background

A *Markov decision process* (MDP) is a mathematical model of the environment with which an agent interacts. Formally, it is a tuple $(\mathcal{S}, \mathcal{A}, \mathcal{P}, \mathcal{R}, d_0, \gamma)$. $\mathcal{S}$ is the set of possible states of the environment. $S_t$ is the state of the environment at time $t \in \{0, 1, \dots\}$. $\mathcal{A}$ is the set of actions that an agent interacting with the environment can take. $A_t$ is the action chosen by the agent at time $t$. For notational simplicity, we assume that $\mathcal{A}$ and $\mathcal{S}$ are finite.[1] $\mathcal{P} : \mathcal{S} \times \mathcal{A} \times \mathcal{S} \to [0, 1]$ is the *transition function*, which characterizes the distribution of $S_{t+1}$ given $S_t$ and $A_t$, using the definition $\mathcal{P}(s, a, s') := \Pr(S_{t+1}{=}s'|S_t{=}s, A_t{=}a)$. The reward provided to the agent at time $t$ is a bounded real-valued random variable, $R_t$. The *reward function* $R : \mathcal{S} \times \mathcal{A} \to [R_{\min}, R_{\max}]$ captures sufficient information about the distribution of rewards given $S_t$ and $A_t$ in order to reason about optimal behavior, and is defined by $R(s, a) := \mathbf{E}[R_t|S_t{=}s, A_t{=}a]$. The initial distribution of states is captured by $d_0 : \mathcal{S} \to [0, 1]$, i.e., $d_0(s) := \Pr(S_0{=}s)$. Finally, $\gamma \in [0, 1]$ is a parameter used to discount rewards based on the time at which they occur.

We consider *episodic* MDPs, which contain a special state $s_\infty$, called the *terminal absorbing state*. Once the agent enters state $s_\infty$, it can never leave and all subsequent rewards are zero. Upon reaching $s_\infty$, the trial, called an *episode*, has effectively ended because there are no more rewards to be obtained. Although in theory the agent will continue transitioning from $s_\infty$ back to $s_\infty$ forever, in practice, we can begin the next episode. We say that a problem has a *finite horizon*, $\tau$, if $S_\tau = s_\infty$ almost surely, regardless of how actions are chosen. A *trajectory* $H$ is the sequence of states, actions, and rewards from one episode: $H = (S_0, A_0, R_0, \dots, S_{\tau-1}, A_{\tau-1}, R_{\tau-1})$.

The mechanism for selecting actions within an agent is called a *policy*, which we denote by $\pi : \mathcal{S} \times \mathcal{A} \to [0, 1]$, where $\pi(s, a) := \Pr(A_t{=}a|S_t{=}s)$. We consider the batch RL setting wherein training data $D$, also referred to as the *safety data*, consists of trajectories generated using a single baseline policy $\pi_b$, called the *behavior policy*. The training data $D$, consists of $n$ trajectories generated using $\pi_b$: $D := \{H_i\}_{i=1}^n$. We write $S_t^i, A_t^i$, and $R_t^i$ to denote the state, action, and reward at time $t$ in the $i^{\text{th}}$ trajectory in $D$. If $\mathcal{H}$ denotes the set of all possible trajectories, each policy induces a distribution over $\mathcal{H}$. We abuse notation by reusing $\pi$, and write $\text{supp}(\pi_b)$ to denote the support of this distribution. Let $\mathcal{H}_{\pi_b} = \text{supp}(\pi_b)$, i.e., the set of all trajectories that can be created by running $\pi_b$.

The *return* is the discounted sum of rewards, and the return from trajectory $H$ is $g(H) := \sum_{t=0}^{\tau-1} \gamma^t R_t$. The goal of the agent is to find a policy that maximizes the expected return it receives. This objective is captured by the objective function: $J(\pi) := \mathbf{E}[\sum_{t=0}^{\tau-1} \gamma^t R_t|\pi]$, where conditioning on $\pi$ denotes that $A_t \sim \pi(S_t, \cdot)$ for all $t$. To simplify notation later, we assume that $\sum_{t=0}^{\tau-1} \gamma^t R_t \in [0, 1]$.[2]

### 2.1  Safe Reinforcement Learning

Let $a$ be a function that takes a dataset as input and produces a policy as output, i.e., $a(D)$ is the policy output by the algorithm $a$ when run on training data $D$. Given a user-specified constant $\delta \in [0, 1]$

(typically $\delta = 0.05$ or $\delta = 0.01$), a *Seldonian* RL algorithm is any algorithm $a$ that satisfies

$$\Pr(J(a(D)) \geq J(\pi_b)) \geq 1 - \delta. \tag{1}$$

That is, the algorithm guarantees that with probability at least $1 - \delta$, it will return a policy with expected return at least equal to that of the behavior policy. For simplicity, we assume that $J(\pi_b)$ is known—in previous papers, $J(\pi_b)$, written instead as $\rho(\pi_b)$, was left as a user-specified constant, for example a high-confidence upper bound on $J(\pi_b)$. Note that an algorithm that always returns $\pi_b$ is technically safe. However, our goal is to develop safe algorithms that frequently return policies that have larger expected return than $\pi_b$, while satisfying the safety guarantee in (1). In this framework, a user can define many different reward functions that improvement can be guaranteed with respect to (w.r.t.). This enables users to define safety constraints using reward functions. For discussion of how this provides a useful interface for defining safety constraints, see the work of Thomas et al. [44].

Several Seldonian RL algorithms have a component called the *safety test*, which ensures that (1) is satisfied [31; 43; 44]. The safety test takes three inputs: 1) A policy $\pi_e$ that is evaluated for safety, referred to as the *evaluation policy*; 2) The safety data, $D$; and 3) $J(\pi_b)$. If there is sufficient confidence that $J(\pi_e) \geq J(\pi_b)$, the safety test returns `True`; otherwise, it returns `False`.

First, using each trajectory in $D$, the safety test computes $n$ estimates of the expected performance of $\pi_e$. For this estimation, we make the standard assumption that $\pi_b(s, a) = 0$ implies $\pi_e(s, a) = 0$.[3] One method for estimating the expected value of a function when samples come from a different distribution ($\pi_b$) than the desired distribution ($\pi_e$) is importance sampling. In Seldonian RL, the safety test uses IS to produce an unbiased estimator of $J(\pi_e)$ from $D$ [38]. Specifically, for each trajectory in $D$, it computes an *importance weighted return* that is defined by: $\hat{J}^\star(\pi_e|H_i, \pi_b) := g(H_i)w^\star(H_i, \pi_e, \pi_b)$, where $w^\star(H_i, \pi_e, \pi_b)$ is the *importance weight*. For traditional IS, $w^{\text{IS}}(H_i, \pi_e, \pi_b) := \prod_{t=0}^{\tau-1} \pi_e(A_t^i, S_t^i)/\pi_b(A_t^i, S_t^i)$. For *weighted importance sampling* (WIS) [27], $w^{\text{WIS}}(H_i, \pi_e, \pi_b) := n(\sum_{x=1}^{n} \prod_{t=0}^{\tau-1} \pi_e(A_t^x, S_t^x)/\pi_b(A_t^x, S_t^x))^{-1} \times \prod_{t=0}^{\tau-1} \pi_e(A_t^i, S_t^i)/\pi_b(A_t^i, S_t^i)$. Notice that WIS normalizes the importance weights using the sum of the importance weights in all trajectories. Given that $\pi_b(s, a) = 0$ implies $\pi_e(s, a) = 0$, IS is strongly consistent and unbiased, while WIS is strongly consistent, but typically biased [42]. For the remainder of the paper, we use $\star \in \{\text{IS}, \text{WIS}\}$ to denote weights computed using IS or WIS.

Second, the safety test uses concentration inequalities to bound the expectation of the importance weighted returns. A CI provides probability bounds on how a random variable deviates from its expectation. Let $\mathbf{X}$ denote the importance weighted returns obtained from the trajectories in $D$, i.e., $\mathbf{X} := \{X_i : i \in \{1, \ldots, n\}, X_i = \hat{J}^\star(\pi_e|H_i, \pi_b)\}$. Safety tests leverage CIs to lower bound the performance of $\pi_e$, using $\mathbf{X}$. A commonly used CI is the *Chernoff-Hoeffding* (CH) inequality [13], which states the following: For $n$ independent, real-valued, and bounded random variables, $X_1, \ldots, X_n$, such that $\Pr(X_i \in [0, b]) = 1$ and $\mathbf{E}[X_i] = \mu$, for all $i \in \{1, \ldots, n\}$, where $b \in \mathbb{R}$, with probability at least $1 - \delta$, $\mu \geq 1/n \sum_{i=1}^{n} X_i - b\sqrt{\ln(1/\delta)/2n}$.

Let $L^{*,\star}(\pi_e, D)$ denote the $1 - \delta$ confidence lower bound on $J(\pi_e)$, calculated using weighting scheme $\star$ and CI $*$, where $* \in \{\text{CH}\}$. Let $\varphi$ denote a safety test such that $\varphi(\pi_e, D, J(\pi_b)) \in \{\text{True}, \text{False}\}$. Given $\pi_e$ and data $D$ collected from $\pi_b$, $\varphi$ returns `True` (i.e., the safety test passes) if $L^{*,\star}(\pi_e, D) \geq J(\pi_b)$; otherwise, $\varphi$ returns `False`. For brevity, we use $L^{*,\star}$ to denote the $1 - \delta$ confidence lower bound on $J(\pi_e)$, calculated using any dataset.

## 3 Related Work

Our paper arguably falls into the broad body of work aimed at creating algorithms that can withstand uncertainty introduced at any stage of the RL framework. Some works view a component as an adversary with stochastic behavior. For an overview of risk-averse methods, e.g., those that incorporate stochasticity into the system, refer to the work of Garcia and Fernandez [7].

Other works incorporate an adversary to model worst-case scenarios. In a *model-free* setting, Morimoto and Doya [32] introduced *Robust RL* that models the environment as an adversary to address parameter uncertainty in MDPs. Pinto et al. [37] extended this work to non-linear policies

that are represented as neural networks. Lim et al. [23] considered MDPs with some adversarial state-action pairs. In *model-based* settings, i.e., those that build an explicit model of the system, although an adversary is not present, worst-case analyses assume different components of an MDP are unknown [34; 46; 39]. Using the definition of safety we have introduced, Ghavamzadeh et al. [8] and Laroche et al. [22] created algorithms for learning safe policies in a model-based setting. Although we are also interested in ensuring security for these approaches, we focus on a model-free setting that requires a different set of assumptions and attacker model.

Learning in the presence of an adversary has also been studied in multi-agent RL, where agents have competing goals [36; 40; 24]. Similar to our work, there have been efforts to study the affect of adversarial inputs to algorithms, and create systems that are "robust" to adversarial manipulation, in multi-armed bandits [17; 25; 12; 49; 48] and on image related tasks in deep RL [15; 19; 5]. To our knowledge, there has not been any research on analyzing adversarial attacks on Seldonian RL.

Outside RL, to increase the robustness of supervised and semi-supervised learning, methods have been proposed during training to play adversarial games [6; 9], and to generate adversarial examples [10].

# 4    Problem Formulation

We focus on a worst-case setting, where an attacker modifies training data to maximize the probability that the Seldonian RL algorithm returns an unsafe policy. When the stakes are high—for example, in the application of RL to sepsis treatment in the ICU, wherein training data is generated from hand-written doctors' notes—we do not want to assume that training data contains only minor errors, such as patient height, but also major ones, such as wrong drug or patient name.

Specifically, we will consider the case where the attacker can add $k$ fabricated trajectories to the safety data $D$. This is related to the case where the attacker can change $k$ of the trajectories in the data—in that setting, the attacker would identify $k$ trajectories to change, and then would replace them with $k$ new ones. After the attacker has identified the $k$ trajectories to change, they are faced with the problem that we solve: which $k$ trajectories to add after the original $k$ were removed.

There are two ways for the safety test to fail: 1) If $J(\pi_e) \geq J(\pi_b)$, the attacker can minimize the estimated performance of $\pi_e$ by adding $k$ trajectories; 2) If $J(\pi_e) < J(\pi_b)$, the attacker can maximize the estimated performance of $\pi_e$ by adding $k$ trajectories. We focus on the latter attacker because the prior case does not cause a poor policy to pass the safety test, but only prevents the identification of a good policy. Therefore, we only consider an attacker who is interested in enabling a worse policy than the behavior policy to pass the safety test. The attacker has the same knowledge inputted to $\varphi$, which consists of: 1) The behavior policy, $\pi_b$, that generated $D$; 2) $J(\pi_b)$; 3) The evaluation policy, $\pi_e$; 4) The CI, $*$, used by $\varphi$; 5) The method to compute importance weights, $\star$; 6) The confidence level, $\delta$; 7) The dataset, $D$.

Let $\mathcal{D}_n^{\pi_b} = \{\mathcal{D} : \mathcal{D} \subseteq \mathcal{H}_{\pi_b} \text{ and } |\mathcal{D}| = n\}$, i.e., the set of all possible datasets of size $n$, created by running $\pi_b$. For all $k \in \mathbb{Z}^+$, let $m : \mathcal{D}_n^{\pi_b} \times \mathbb{Z}^+ \to \mathcal{D}_{n+k}^{\pi_b}$ be the *attack function*, which indicates a strategy that an attacker might use when appending fabricated trajectories to the dataset. That is, for $D \in \mathcal{D}_n^{\pi_b}$, $m(D, k) \in \mathcal{D}_{n+k}^{\pi_b}$ is the dataset created by using attack strategy $m$ to append $k$ trajectories to $D$ with size $n$. Let $\mathcal{M}$ denote the set of all possible attack functions $m$. For notational completeness, we note that $m$ is actually a function of $\mathcal{D}_n^{\pi_b}$, $\mathbb{Z}^+$ and items 1–6 enumerated in the previous paragraph. However, we omit these items for brevity.

Next, we define $\alpha$-security, which provides one way of quantifying how robust a safety test (and thus a Seldonian algorithm relying on a safety test) is to adversarial perturbations of data in terms of a parameter, $\alpha \geq 0$, such that smaller values of $\alpha$ correspond to more robustness. We use the following assumptions when defining $\alpha$-security:

**Assumption 1 (Inferior $\boldsymbol{\pi_e}$).** $J(\pi_e) < J(\pi_b)$.

**Assumption 2 (Absolute continuity).** $\forall a \in \mathcal{A}, \forall s \in \mathcal{S}, \big(\pi_b(s, a) = 0\big) \implies \big(\pi_e(s, a) = 0\big)$.

**Assumption 3 ($\boldsymbol{\varphi}$ safety).** *We only consider safety tests that ensure* (1) *is satisfied by any algorithm that returns $\pi_e$ if $\varphi\big(\pi_e, D, J(\pi_b)\big) = \texttt{True}$, and $\pi_b$ otherwise. That is, given Assumption 1, $\varphi$ must satisfy* $\Pr(\varphi(\pi_e, D, J(\pi_b)) = \texttt{True}) < \delta$.

Recall that the attacker is interested in enabling a worse policy than $\pi_b$ to pass the safety test by appending trajectories to $D$. In order to succeed, they must manipulate the metric used for decision making by the safety test, i.e., artificially increase $L^{*,\star}$, the $1 - \delta$ confidence lower bound on $J(\pi_e)$.

Our definition ensures (1) is satisfied even if $D$ has been corrupted. In fact, an undesirable policy passing the safety test with probability at most $\delta$, must hold across all attack functions, which includes the best attack strategy—one that causes the largest increase in $L^{*,\star}$. This artificial increase in $L^{*,\star}$, due to the best attack strategy, is represented by $\alpha$, which we write as a function of $n$, $k$, $\pi_b$, $\pi_e$ and $\delta$.

**Definition 1** ($\boldsymbol{\alpha}$**-security**)**.** *Under Assumptions 1, 2 and 3, $\varphi$ is secure with constant $\alpha$ for $\pi_e$, $\pi_b$, $k$, and $D$ collected from $\pi_b$, where $|D| = n$, if and only if,*

$$\forall m \in \mathcal{M}, \Pr\Big(\varphi\big(\pi_e, m(D, k), J(\pi_b) + \alpha\big) = \texttt{True}\Big) < \delta.$$

If a safety test does not satisfy Assumption 3, we can not easily analyze its $\alpha$-security. For example, any $\varphi$ using WIS variants does not satisfy Assumption 3 because many CIs often rely on the independence of samples to guarantee probabilistic bounds on their mean. WIS creates dependence between samples by normalizing the importance weights. However, WIS variants are more likely to be used by safety tests because they work better in practice and require less data than IS. Therefore, to include WIS in our analysis, we define quasi-$\alpha$-security to be the following:

**Definition 2** (**Quasi-$\boldsymbol{\alpha}$-security**)**.** *Under Assumptions 1 and 2, $\varphi$ is quasi-secure with constant $\alpha$ for $\pi_e$, $\pi_b$, $k$, and $D$ collected from $\pi_b$, where $|D| = n$, if and only if,*

$$\forall m \in \mathcal{M}, \Pr\Big(\varphi\big(\pi_e, m(D, k), J(\pi_b) + \alpha\big) = \texttt{True}\Big) \leq \Pr\Big(\varphi\big(\pi_e, D, J(\pi_b)\big) = \texttt{True}\Big). \quad (2)$$

Note that (2) implies $\alpha$-security if $\varphi$ is "safe". If it is not (as in WIS variants), then $\varphi$ can be quasi-$\alpha$-secure, which still gives us a way to measure its robustness to perturbations/attacks on data.

Notice that our definition does not consider how $\pi_e$ is chosen because the violation of safety comes primarily from the safety test. In addition to the attacker model, our worst-case analysis also stems from this definition of security, which assumes that $\pi_e$ has lower performance than $\pi_b$, but does not quantify how often this occurs. Attacking the data used to select $\pi_e$ can increase this frequency, but would not change the definition of $\alpha$-security.

## 5 Analysis of Existing Algorithms

In this section, we present our main contribution which quantifies the $\alpha$-security of different off-policy performance estimators. Notice that every safety test is $\alpha$-secure with $\alpha = \infty$, since the test whether $L^{*,\star} > J(\pi_b) + \infty$ will always return `False`. So, when comparing two estimators, it is not sufficient to compare arbitrary values of $\alpha$ for which they are $\alpha$-secure. Instead, we define the notion of a *minimum* $\alpha$, which we refer to as $\alpha^*$:

**Definition 3** (**Tight $\boldsymbol{\alpha}$**)**.** *$\varphi$ is quasi-secure or secure with tight constant $\alpha^*$ if and only if $\alpha^* = min\{\alpha : \varphi \text{ is quasi-}\alpha\text{-secure or }\alpha\text{-secure}\}$.*

The value of $\alpha$ can be interpreted as the largest possible increase in $L^{*,\star}$ when an attacker adds $k$ trajectories to $D$. To compute $\alpha^*$, given a random dataset, we first must determine the optimal attack strategy, where this increase is largest. This strategy can be determined using the following realization: $L^{\text{CH, IS}}$ and $L^{\text{CH, WIS}}$ are both increasing functions w.r.t. the IS weight and return. Therefore, for a given $k$, the optimal attack is to create a trajectory that maximizes the value of the IS weight and return. This strategy incurs the maximum "damage" by the attacker.

Notice that fake trajectories created using this strategy are those that have not been performed in the real MDP environment. However, because the transition and reward functions are not known, a practitioner can not distinguish real and fake trajectories. Rare events are critical to account for in RL, and may look like fake trajectories. Perhaps impossible trajectories can be identified using domain-specific knowledge, but that must be analyzed on a per-domain basis.

Second, computing $\alpha^*$ requires identifying the dataset on which the attack strategy is most effective, i.e., the determination of $D \in \mathcal{D}_n^{\pi_b}$ that yields the greatest increase in $L^{*,\star}$. However, this is not possible since the distribution of trajectories for a given $\pi_e$ is unknown, and thus, $\mathcal{D}_n^{\pi_b}$ is unknown.

Table 1: **$\alpha$–security of current methods (center); settings for clipping weight, $c$, for $\alpha$-security written in terms of a user-specified $k$ and $\alpha$ (right).** The minimum IS weight is denoted by $i^{\min}$.

| Estimator | $\alpha'$ | $c$ |
|---|---|---|
| CH, IS | $i^*\left(\sqrt{\frac{\ln(1/\delta)}{2n}} - \sqrt{\frac{\ln(1/\delta)}{2(n+k)}} + \frac{k}{(n+k)}\right)$ | $\dfrac{\alpha}{\left(\sqrt{\frac{\ln(1/\delta)}{2n}} - \sqrt{\frac{\ln(1/\delta)}{2(n+k)}} + \frac{k}{(n+k)}\right)}$ |
| CH, WIS | $\sqrt{\frac{\ln(1/\delta)}{2n}} - \sqrt{\frac{\ln(1/\delta)}{2(n+k)}} + \frac{ki^*}{(i^{\min}+ki^*)}$ | $\dfrac{i^{\min}\left(\alpha - \sqrt{\frac{\ln(1/\delta)}{2n}} + \sqrt{\frac{\ln(1/\delta)}{2(n+k)}}\right)}{k\left(1-\alpha+\sqrt{\frac{\ln(1/\delta)}{2n}} - \sqrt{\frac{\ln(1/\delta)}{2(n+k)}}\right)}$ |

Instead, we propose the use of a different value, $\alpha'$, which may be slightly loose relative to the tight $\alpha^*$, but which we expect captures the relative robustness of the methods that we compare. Instead of $D \in \mathcal{D}_n^{\pi_b}$, $D \in \mathcal{D}_n^{\mathcal{H}}$ is selected, on which the attacker executes the optimal strategy. In other words, the dataset is chosen out of all datasets of size $n$ created by $\mathcal{H}$, the set of all trajectories that can be created by any policy. In Theorem 1, we present the values of $\alpha'$ for each estimator.

**Theorem 1.** $\varphi$ *is quasi-secure or secure with $\alpha \geq \alpha'$, where the values of $\alpha'$ are presented in Table 1.*

*Proof.* In Appendix A, we prove that $L^{\text{CH, IS}}$ and $L^{\text{CH, WIS}}$ are increasing functions w.r.t. the IS weight and return. The largest IS weight is a function of $\pi_b$, $\pi_e$ and $\tau$, and the largest return is always 1. Let any off-policy performance estimator, $L^{*,\star}(\pi_e, D)$, be written as a function $f$ that incorporates an attacker strategy. In other words, $L^{*,\star}(\pi_e, m(D, k)) = f^{*,\star}(D, w_y, g_y, k)$, where $w_y$ is the IS weight and $g_y$ is the return, computed from the trajectory added by the attacker. That is, $f^{*,\star}(D, w_y, g_y, k)$ is the result of applying CI $*$ and weighting scheme $\star$ to $D$ that includes $k$ copies of $H^*$, which is the trajectory added by the attacker. $H^*$ has an IS weight of $w_y$ and return of $g_y$.

The optimal attack strategy causes the largest increase in $L^{*,\star}$ such that $L^{*,\star}(\pi_e, m(D, k)) - L^{*,\star}(\pi_e, D)$ is maximized. In Lemma 5 in Appendix A, we prove that an appropriate setting of $\alpha$ is equal to or greater than the largest increase in $L^{*,\star}$ across all datasets, $D \in \mathcal{D}_n^{\pi_b}$, and all attack functions. By Lemma 1, a safety test using $L^{*,\star}$ is quasi-$\alpha$-secure or $\alpha$-secure if $\forall D \in \mathcal{D}_n^{\pi_b}$ and $\forall m \in \mathcal{M}$,

$$\alpha \geq L^{*,\star}\big(\pi_e, m(D, k)\big) - L^{*,\star}(\pi_e, D). \tag{3}$$

Let $U = \{u : \exists D \in \mathcal{D}_n^{\pi_b}, \exists m \in \mathcal{M}, u = L^{*,\star}\big(\pi_e, m(D, k)\big) - L^{*,\star}(\pi_e, D)\}$, i.e., permissible values of $\alpha^*$ obtained from each $D \in \mathcal{D}_n^{\pi_b}$. Let $\alpha^* = \max_{D \in \mathcal{D}_n^{\pi_b}} \max_{m \in \mathcal{M}} L^{*,\star}\big(\pi_e, m(D, k)\big) - L^{*,\star}(\pi_e, D)$. For all $u \in U$,

$$u \leq \max_{D \in \mathcal{D}_n^{\pi_b}} \max_{m \in \mathcal{M}} L^{*,\star}\big(\pi_e, m(D, k)\big) - L^{*,\star}(\pi_e, D)$$
$$= \max_{D \in \mathcal{D}_n^{\pi_b}} f^{*,\star}(D, i^*, 1, k) - L^{*,\star}(\pi_e, D). \tag{4}$$

Besides the optimal attacker strategy, if the distribution of trajectories for a given $\pi_e$ was also known, the right-hand side of (4), which is $\alpha^*$, would be computable. Instead, an upper bound of $\alpha^*$ is

$$\alpha^* < \max_{D \in \mathcal{D}_n^{\mathcal{H}}} f^{*,\star}(D, i^*, 1, k) - L^{*,\star}(\pi_e, D) = \alpha', \tag{5}$$

where $\mathcal{D}_n^{\mathcal{H}} = \{\mathcal{D} : \mathcal{D} \subseteq \mathcal{H} \text{ and } |\mathcal{D}| = n\}$. Substituting the definition of $f^{*,\star}$ and $L^{*,\star}$ into the right-hand side of (5) for various weighting schemes and CIs, we solve for a clean expression for $\alpha'$, presented in Table 1. For algebraic details, refer to Appendix B. This upper bound of $\alpha^*$, $\alpha'$, satisfies (3), implying that any $\alpha$ such that $\alpha \geq \alpha'$ also satisfies (3). □

Recall the following to interpret Table 1: 1) The IS weight is a ratio of the probability of observing a trajectory under $\pi_e$ to that of $\pi_b$; 2) We only consider MDPs with finite length, $\tau$. The largest IS weight, $i^*$, is a function of $\pi_e$, $\pi_b$ and $\tau$. To compute $i^*$: 1) Through brute search, for a single time step, a state and action pair is selected such that the ratio of its probability under $\pi_e$ to $\pi_b$ is maximized; 2) If this ratio is greater than 1, the pair is repeated for the length of the trajectory, exponentially increasing the IS weight. Notice that as $i^{\min} \to 0$, the third term of $\alpha'$ for WIS equals 1. Plus, due to the normalization of IS weights, importance weighted returns are always bounded by 1

for WIS. Therefore, $\alpha'$ for WIS is much smaller than that of IS, and in the worst-case, is slightly over 1. The $\alpha'$ for IS is roughly the same as that of WIS, but scaled by $i^*$ that can be massive.

The conclusion to draw is not that WIS is more robust than IS because $\alpha$-security does not capture the whole story. The difference in true performance of $\pi_e$ and $\pi_b$, in addition to how accurate $L^{\text{CH},*}$ estimates the performance of $\pi_e$ on a given uncorrupted dataset, both contribute to the robustness of the estimators to data anomalies. Yet $\alpha$-security is useful for deciding whether $\pi_e$ is worth evaluating for a given estimator. As $\alpha'$ increases, $L^{*,\star}$ can artificially be increased by a greater amount; hence, given the choice of two evaluation policies, a practitioner might pick the policy with lower $\alpha'$. If $\alpha' = 0$, $\varphi$ for $\pi_e$ is highly robust against an adversary.

Although we focus on Chernoff-Hoeffding, our analysis also extends to other bounds such as Azuma [2] and Bernstein [30]. Moreover, *per-decision importance sampling* and *weighted per-decision importance sampling* are equivalent to IS and WIS, respectively, when rewards are at the end of a trajectory. So, our analysis applies to these specific cases as well.

## 6 Panacea: An Algorithm for Safe and Secure Policy Improvement

In this section, we describe our algorithm `Panacea`, named after the Greek goddess of healing, that provides $\alpha$-security, with a user-specified $\alpha$, if the number of corrupt trajectories in $D$ is upper bounded. That is, the important additional input to the algorithm is the number of adversarial trajectories. The algorithm also takes as input all the information that the attacker already knows; except, is agnostic to *which* or *how many* of the trajectories in $m(D, k)$ have been corrupted.

`Panacea` caps the importance weights using some clipping weight, $c$. First introduced for RL by Bottou et al. [4], clipping works by computing all the IS weights in $D$ and capping the weights to $c$, i.e., if any IS weight is greater than $c$, it is set to $c$. Given a user-specified $\alpha$ and $k$ as input, `Panacea` computes $c$, using the values in Table 1. It then creates a clipped version of the dataset, denoted by `Panacea`$(D, c)$. For pseudocode, refer to Algorithm 1 in Appendix C. In Corollary 1, we show that when $k$ is chosen correctly, our algorithm meets a user-specified level of security against an attacker, whose optimal strategy does not change even if they know we are using `Panacea`. In Corollary 2, we discuss how the clipping weights are computed.

**Corollary 1.** *If the user upper bounds $k$, Panacea is quasi-secure or secure with $\alpha \geq \alpha'$, where $\alpha'$ is user-specified.*

*Proof.* The behavior of $L^{*,\star}$ does not change with `Panacea`: $L^{*,\star}$ is increasing w.r.t. the IS weight and return, regardless of their range of values. So, the optimal attacker strategy remains the same. Let $k$ denote the number of adversarial trajectories added by the attacker, and $k'$ denote the input provided to `Panacea` by the user. When this value is upper bounded correctly, $k' = k$, and `Panacea` computes a clipping weight for the given estimator, denoted by $c^{*,\star}$, using Table 1. By Theorem 1, the security of `Panacea` is

$$\max_{D \in \mathcal{D}_n^{\mathcal{H}}} \max_{m \in \mathcal{M}} L^{*,\star}\big(\pi_e, \texttt{Panacea}(m(D,k), c^{*,\star})\big) - L^{*,\star}\big(\pi_e, \texttt{Panacea}(D, c^{*,\star})\big)$$

$$= \max_{D \in \mathcal{D}_n^{\mathcal{H}}} f^{*,\star}(\texttt{Panacea}(D, c^{*,\star}), c^{*,\star}, 1, k) - L^{*,\star}\big(\pi_e, \texttt{Panacea}(D, c^{*,\star})\big) \quad (6)$$

$$\leq \alpha'.$$

In Appendix C.1, we solve for a clean expression of (6) by substituting in the definition of $f^{*,\star}$, $L^{*,\star}$ and $c^{*,\star}$ for various weighting schemes and CIs, and then simplifying the expression. $\square$

**Corollary 2.** *If the user upper bounds $k$, Panacea is quasi-$\alpha$-secure or $\alpha$-secure with the values of $c$ in Table 1.*

*Proof.* If the number of adversarial trajectories in $D$ is upper bounded, rewriting (6) in terms of $c^{*,\star}$, and then solving (6) for a clean expression for $c^{*,\star}$ by substituting $\alpha'$, $k$ and $L^{*,\star}$ with the user-specified inputs, equals the clipping weights found in Table 1 for different estimators. For algebraic details, refer to Appendix C.2. $\square$

`Panacea` is more secure than existing methods if the user correctly determines $k$ and selects a value of $\alpha$ that is less than that of existing methods. Only then is $c < i^*$, and `Panacea` clips the $k$ largest IS

weights in $m(D, k)$ to $c$. Additionally, if $c < I_n$, where $I_n$ is the largest IS weight in the uncorrupted dataset $D$, then some values in $D$ are also clipped. Table 1 in Appendix C is the same as the middle column of Table 1 with one modification: $i^*$ are replaced with $c$. As $c$ decreases, more values are collapsed, $L^{\text{CH},\star}$ is lower, and the probability of returning $\pi_b$ is higher.

# 7  Empirical Evaluation

We quantify the $\alpha$-security of two safety tests, with and without `Panacea`, applied to two domains: a grid-world, where the deployment of an unsafe policy has low-stakes, and a diabetes treatment simulation, where the deployment of an unsafe policy could lead to very dangerous outcomes.

## 7.1  Experimental Methodology and Application Domains

**Grid-world.** In a classic $3 \times 3$ grid-world, the agent's goal is to reach the bottom-right corner of the grid, starting from the top-left corner. When viewed as an MDP, actions correspond to directions the agent can move, and states correspond to its current location on the grid. The agent receives a reward of 1, discounted by $\gamma^t$, if they reach the bottom-right corner; otherwise all rewards are 0.

**Diabetes Treatment Simulation.** For the diabetes treatment simulation, we use a Python implementation [47] of an FDA-approved type 1 diabetes Mellitus simulator (T1DMS) by Kovatchev et al. [20] and Man et al. [28]. The simulator simulates the metabolism of a patient with type 1 diabetes, where the body does not make enough insulin, a hormone needed for moving glucose into cells. Insulin pumps have a bolus calculator that determines how much insulin, specifically known as bolus insulin, must be injected into the body before having a meal. One type of bolus calculator is parametrized by two real-valued parameters, CR and CF. The MDP view of the simulator represents the bolus calculator as a policy, injection doses as actions, and states as the patient's body's reactions to consuming meals and getting insulin injections. We adopt a similar reward function used in previous work that penalizes any deviation from optimum levels of blood glucose [3].

**Data Collection.** We use RL to search the space of policies for grid-world, and the space of probability distributions over policies for the diabetes domain [44]. For the latter case, we assign the mode of a triangular distribution to a value sampled uniformly from range $[5, 50]$ for CR and $[1, 31]$ for CF—admissible ranges for any diabetic patient. Also, both triangular distributions have the same range from which CR and CF values are sampled. The two modes parametrize the policy space over which we sample policies. $D$ is created by sampling a CR and CF pair from their respective distributions, and observing the return. This reformulation is a bandit problem, where the action corresponds to picking the modes of two triangular distributions from which to sample CR and CF.

Let $\pi(v, u, \theta_1, \theta_2)$ denote a policy representing the joint probability of sampling $v$ under a triangular distribution with mode $\theta_1$ and range $[5, 50]$, and sampling $u$ under a triangular distribution with mode $\theta_2$ and range $[1, 31]$. Instead of finding a trajectory with the largest IS weight and return, the optimal attack strategy is selecting a CR$'$ and CF$'$ such that the ratio of their joint probability under $\pi_e$ to that of $\pi_b$ is maximized, i.e., $\arg\max_{\text{CR}' \in [5,50]} \arg\max_{\text{CF}' \in [1,31]} \pi_e(\text{CR}', \text{CF}', \theta_3, \theta_4) / \pi_b(\text{CR}', \text{CF}', \theta_1, \theta_2)$. The attacker then adds $k$ copies of CR$'$ and CF$'$, along with a return of 1, to $D$. Our results show that corrupting $D$ collected from adult#003 within T1DMS can cause a Seldonian RL algorithm to select a bad policy, i.e., a new distribution over policies with lower return than $\pi_b$.

## 7.2  Results and Discussion

We evaluated the number of trajectories that must to be added to $D$ before $\varphi$ incorrectly returns unsafe policies with probability more than $\delta$. The goal is not to show how much data it takes to "break" each method—the goal is to show that, without `Panacea`, both methods are extremely fragile. For our experimental setup, we selected two policies per domain. We estimated $J(\pi_b) \approx 0.797$ and $J(\pi_e) \approx 0.728$ for grid-world, and $J(\pi_b) \approx 0.218$ and $J(\pi_e) \approx 0.145$ for the diabetes domain, by averaging returns obtained from running each policy 10,000 times. We added $k$ adversarial trajectories based on the optimal attacker strategy to a randomly created $D$ of size 1,500. We executed the safety test by comparing $J(\pi_b)$ to $L^{\text{CH},\star}$, computed using the corrupt data. Figure 1 shows the average $L^{\text{CH},\star}$ over 750 trials, as $k$ increases. The error bars for variance are so small that they are negligible. Although not shown, the average probability of passing the safety test across all trials is around 0% before and 100% after the blue and red lines cross the black dotted line, representing $J(\pi_b)$.

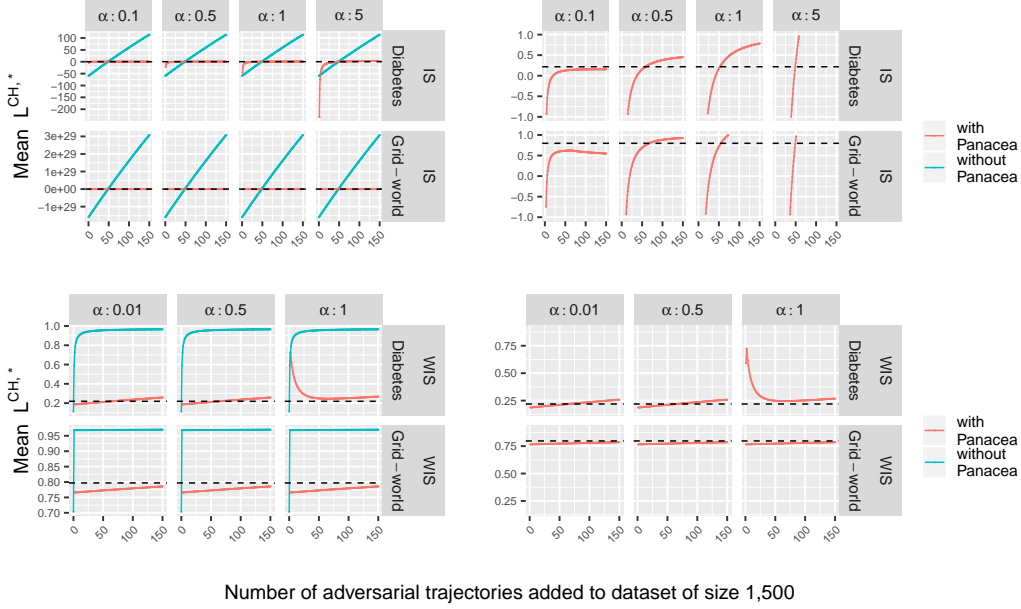

Number of adversarial trajectories added to dataset of size 1,500

Figure 1: **Mean $f^{\text{CH}, *}$ as $k$ increases with magnified view (right).** Facets represent the value of $\alpha$ inputted to Panacea, and do not matter for existing methods, labelled as "without Panacea". For IS, Panacea crosses the black dotted line at $k = 59, 53$ and $49$ for the diabetes domain, and $k = 68, 55$ and $50$ for grid-world, when $\alpha = 0.5, 1$ and $5$, respectively. For WIS, Panacea crosses the black dotted line at $k = 65, 65$ and $1$ for the diabetes domain, when $\alpha = 0.01, 0.5$ and $1$, respectively. Panacea, using WIS, does not cross the black dotted line for grid-world.

The results without Panacea show that $L^{\text{CH, IS}}$ and $L^{\text{CH, WIS}}$ cross the black dotted line at $k = 49$ and $k = 1$, respectively, for both domains. For WIS, it only takes a single trajectory for $\varphi$ to return an unsafe policy; for IS, 49 trajectories only constitute $3.2\%$ of $D$, where $|D| = 1,549$. This could correspond to a morning's worth of data collected from an incorrectly parametrized insulin pump, or the temporary period over which the sensors of a self-driving car are not working due to severe weather conditions. Notice that these results are not indicative of IS being less sensitive than WIS: Because WIS estimates performance more accurately, small changes in its estimate cause the incorrect conclusion that $J(\pi_e) > J(\pi_b)$.

For Panacea, we computed the clipping weights, found in Table 1 per estimator, using the user-specified $\alpha$ and the actual number of adversarial trajectories added by the attacker. Our method never crosses the black dotted line for either domain at $\alpha = 0.1$ with IS; and it requires 65 adversarial trajectories to break rather than a single trajectory for the diabetes domain at $\alpha = 0.01$ with WIS. Notice how the parameter settings affect $\alpha$-security: 1) As $\alpha$ increases, so does the clipping weight. 2) As $k$ increases, the clipping weight decreases to counteract the artificial increase of $L^{\text{CH},\star}$ due to corrupt trajectories. In the worst-case—when the user inputs an $\alpha$ that is greater than the $\alpha'$ of existing methods—Panacea requires the same number of adversarial trajectories to break as if not used because $D$ is not clipped at all.

One of the practical considerations when deploying Panacea is estimating the number of corrupt trajectories in training data. For areas such as natural language processing and computer vision, a user might address this concern by using known error rates in the data processing pipelines of well-known models. Also, choosing a meaningful value for $\alpha$ might be challenging. Domain-specific knowledge of the range of performance for different policies—especially the difference in performance between good and bad policies—can be useful. But notice that WIS estimates are always bounded by 1, and in practice, IS estimates are even smaller—when computing the IS weight, the probability of a trajectory under $\pi_e$ is often much smaller than that of $\pi_b$. The middle column in Table 1 is usually $\geq 1$, indicating that standard methods can be completely broken (make pessimal policies appear optimal) easily. The right column shows values of $c$ that make Panacea $\alpha$-secure for *any* $\alpha \in [0, \infty]$. However, plugging in $\alpha \in [0, 1]$ such that $\alpha \leq \alpha'$ gives Panacea a meaningful security guarantee.

## Broader Impact

In our paper, we discussed the application of Seldonian algorithms to the treatment of diabetes patients. We emphasize that the mathematical safety guarantees provided by Seldonian RL are not a replacement for domain-specific safety requirements (e.g., the diabetes treatment would still need oversight for medical safety), but still improve the potential for RL to be applied to problems with real-world consequences. Seldonian RL has also been proposed for creating fair algorithms, i.e., those that aim to reduce discriminative behavior in intelligent tutoring systems and loan approvals [31].

In the last decade, data breaches on the Democratic National Committee's emails, and on companies such as Equifax and Yahoo! have made cyber attacks on systems and databases a very legitimate and ubiquitous concern [45; 33; 14]. Therefore, when creating safe AI algorithms that can directly impact people's lives, we should ensure not only performance guarantees with high probability, but also the development of metrics that evaluate the "quality" of training data, which often reflect systemic biases and human error.

## Acknowledgements

We thank Yash Chandak and Stephen Giguere for helpful discussions. This work was supported in part by NSF Award #2018372 and a gift from Adobe. Research reported in this paper was also sponsored in part by the CCDC Army Research Laboratory under Cooperative Agreement W911NF-17-2-0196 (ARL IoBT CRA). The views and conclusions contained in this document are those of the authors and should not be interpreted as representing the official policies, either expressed or implied, of the Army Research Laboratory or the U.S. Government. The U.S. Government is authorized to reproduce and distribute reprints for Government purposes notwithstanding any copyright notation herein.

## Footnotes

[1]Our work generalizes to continuous MDPs as well, but care must be taken to select $\pi_e$ such that importance sampling weights are bounded.

[2]This is equivalent to assuming that rewards are bounded, since given a finite horizon, the returns can be normalized.

[3]In the more general setting, where $\mathcal{A}$ and $\mathcal{S}$ are not finite, this assumption corresponds to requiring $\pi_e$ to be absolutely continuous with respect to $\pi_b$.

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
