[Supplementary Material]

# A  Analysis of Existing Algorithms

Let $f^{*,\star}$ denote a function that incorporates an attacker strategy. When $k = 0$, $f^{\text{CH, IS}}(D, w_y, g_y, k)$ is the result of applying the CH inequality to the IS weighted returns, obtained from $D$, which additionally includes $k$ copies of a trajectory with an IS weight of $w_y$ and return of $g_y$. Notice that $f^{*,\star}$ is written in terms of IS weights. The following defines $f^{\text{CH, WIS}}$, written in terms of IS weights, when $k = 0$:

$$f^{\text{CH, WIS}}(D, w_y, g_y, 0) = \frac{1}{\sum_{i=1}^n w_i} \sum_{i=1}^n w_i g_i - b \sqrt{\frac{\ln(1/\delta)}{2n}}.$$

For the rest of the paper, we use the following notation. Let $\mathcal{I} = \{I : \exists a \in \mathcal{A}, \exists s \in \mathcal{S}, I = \prod_{t=0}^{\tau-1} \pi_e(A_t = a, S_t = s)/\pi_b(A_t = a, S_t = s)\}$, i.e., the set of all IS weights that could be obtained from policies $\pi_e$ and $\pi_b$. The maximum and minimum IS weight is denoted by $i^* = \max(\mathcal{I})$ and $i^{\min} = \min(\mathcal{I})$, respectively. For shorthand, let the sum of IS weights in $D$ be written as $\beta = \sum_{i=1}^n w_i$. Also, we assume that $\beta > 0$ to ensure that WIS is well-defined.

Next, we define a new term to describe how an attacker can increase the $1 - \delta$ confidence lower bound on the mean of a bounded and real-valued random variable. We say that $f^{*,\star}$ *is adversarially monotonic given its inputs*, if an attacker can maximize $f^{*,\star}$ by maximizing the value of the added samples. For brevity, we say that $f^{*,\star}$ is adversarially monotonic.

**Definition 1.** $f^{*,\star}$ *is adversarially monotonic for $n > 1$, $k > 0$, $\pi_b$, $\pi_e$ and $D$ if both*

1. *There exists two constants $p \geq 0$ and $q \in [0,1]$, with $pq \in [0, i^*]$, such that $f^{*,\star}(D, p, q, k) \geq f^{*,\star}(D, p, q, 0)$, i.e., adding $k$ copies of $pq$ does not decrease $f$;*

2. $\frac{\partial}{\partial g_y} f^{*,\star}(D, i^*, g_y, k) \geq 0$ *and* $\frac{\partial}{\partial w_y} f^{*,\star}(D, w_y, 1, k) \geq 0$, *with no local maximums, i.e., $f$ is a non-decreasing function w.r.t. the IS weight and return added by the attacker, respectively.*

Definition 1 means that $f^{*,\star}$ is maximized when $w_y$ and $g_y$ is maximized. In other words, the optimal strategy is to add $k$ copies of the trajectory with the maximum IS weight and return. Notice that $f^{*,\star}$ does not incorporate all possible attack functions, $\mathcal{M}$: specifically, the set of attacks, where the attacker can choose to add $k$ different trajectories, is omitted. As described in Theorem 1, to perform a worst-case analysis, only the optimal attack must be incorporated as part of $f^{*,\star}$.

In the following two lemmas, we show that a couple well-known Seldonian algorithms are adversarially monotonic.

**Lemma 1.** *Under Assumptions 1, 2 and 3, $f^{\text{CH, IS}}$ is adversarially monotonic.*

*Proof.* Let $w_y \geq \frac{1}{n} \sum_{i=1}^n w_i g_i + \frac{(n+k)}{k} \left( b \sqrt{\frac{\ln(1/\delta)}{2(n+k)}} - b \sqrt{\frac{\ln(1/\delta)}{2n}} \right)$ and $g_y = 1$. To show that $w_y g_y \in [0, i^*]$ as stated in (1) in Definition 1, it must be that $w_y \in [0, i^*]$. For all $i \in \{1, \ldots, n\}$, $w_i g_i \in [0, i^*]$. Thus, for any given dataset, $0 \leq 1/n \sum_{i=1}^n w_i g_i \leq i^*/n$. Using this fact, for any given $D$, the range of $w_y$ is

$$\frac{1}{n} \sum_{i=1}^n (0) + \frac{(n+k)}{k} \left( b \sqrt{\frac{\ln(1/\delta)}{2(n+k)}} - b \sqrt{\frac{\ln(1/\delta)}{2n}} \right) \leq w_y \leq \frac{1}{n} \sum_{i=1}^n (i^*) + \frac{(n+k)}{k} \left( b \sqrt{\frac{\ln(1/\delta)}{2(n+k)}} - b \sqrt{\frac{\ln(1/\delta)}{2n}} \right)$$

$$\underbrace{\frac{b(n+k)}{k} \left( \sqrt{\frac{\ln(1/\delta)}{2(n+k)}} - \sqrt{\frac{\ln(1/\delta)}{2n}} \right)}_{<0} \leq w_y \leq \frac{i^*}{n} + \underbrace{\frac{b(n+k)}{k} \left( \sqrt{\frac{\ln(1/\delta)}{2(n+k)}} - \sqrt{\frac{\ln(1/\delta)}{2n}} \right)}_{<0} \leq i^*.$$

Therefore, $w_y$ can be selected such that $w_y g_y \in [0, i^*]$. It follows that

$$
\begin{aligned}
f^{\text{CH, IS}}(D, w_y, 1, k) =& \frac{1}{n+k} \sum_{i=1}^{n} w_i g_i + \frac{k}{n+k}(w_y)(1) - b\sqrt{\frac{\ln(1/\delta)}{2(n+k)}} \\
\geq & \frac{1}{n+k} \sum_{i=1}^{n} w_i g_i + \frac{k}{n+k} \left( \frac{1}{n} \sum_{i=1}^{n} w_i g_i + \frac{(n+k)}{k} \left( b\sqrt{\frac{\ln(1/\delta)}{2(n+k)}} - b\sqrt{\frac{\ln(1/\delta)}{2n}} \right) \right) - b\sqrt{\frac{\ln(1/\delta)}{2(n+k)}} \\
=& \frac{1}{n} \sum_{i=1}^{n} w_i g_i - b\sqrt{\frac{\ln(1/\delta)}{2n}} \\
=& f^{\text{CH, IS}}(D, w_y, g_y, 0).
\end{aligned}
$$

Next, we show that (2) in Definition 1 holds.

$$
\begin{aligned}
\frac{\partial}{\partial w_y} f^{*,\star}(D, w_y, g_y, k) =& \frac{\partial}{\partial w_y} \left( \sum_{i=1}^{n} \frac{w_i g_i}{n+k} \right) + \frac{k w_y g_y}{n+k} - b\sqrt{\frac{\ln(1/\delta)}{2(n+k)}} \\
=& \frac{k g_y}{n+k} \\
\frac{\partial}{\partial w_y} f^{*,\star}(D, w_y, 1, k) =& \frac{k}{n+k}. \\
\frac{\partial}{\partial g_y} f^{*,\star}(D, w_y, g_y, k) =& \frac{\partial}{\partial g_y} \left( \sum_{i=1}^{n} \frac{w_i g_i}{n+k} \right) + \frac{k w_y g_y}{n+k} - b\sqrt{\frac{\ln(1/\delta)}{2(n+k)}} \\
=& \frac{k w_y}{n+k} \\
\frac{\partial}{\partial g_y} f^{*,\star}(D, i^*, g_y, k) =& \frac{k i^*}{n+k}.
\end{aligned}
$$

Notice that both partial derivatives are non-negative when $g_y = 1$ and $w_y = i^*$, respectively. To find any critical points, the following equations are solved simultaneously: $\partial/\partial g_y f^{\text{CH, WIS}}(D, w_y, g_y, k) = 0$ and $\partial/\partial w_y f^{\text{CH, WIS}}(D, w_y, g_y, k) = 0$. Notice that points along the line $(w_g, 0)$ and $(0, g_y)$ are all critical points. The following partial derivatives are computed to classify these points:

$$
\begin{aligned}
\frac{\partial}{\partial(w_y)^2}(D, w_y, g_y, k) =& 0. \\
\frac{\partial}{\partial(g_y)^2}(D, w_y, g_y, k) =& 0. \\
\frac{\partial}{\partial g_y w_y}(D, w_y, g_y, k) =& \frac{k}{n+k}.
\end{aligned}
$$

Using the second partial derivative test, the critical points are substituted into the following equation:

$$
\frac{\partial}{\partial(w_y)^2} \cdot \frac{\partial}{\partial(g_y)^2} - \left( \frac{\partial}{\partial g_y w_y} \right)^2 = - \left( \frac{k}{n+k} \right)^2,
$$

which is less than zero. Therefore, points along the line $(w_g, 0)$ and $(0, g_y)$ are saddle points. $\quad\square$

**Lemma 2.** *Under Assumptions 1 and 2, $f^{\text{CH, WIS}}$ is adversarially monotonic.*

*Proof.* First, we show that (1) in Definition 1 holds with $g_y = 1$ and $w_y = 0$.

$$f^{\text{CH, WIS}}(D, w_y, g_y, k) = \frac{1}{kw_y + \beta}\left(kw_y g_y + \sum_{i=1}^{n} w_i g_i\right) - b\sqrt{\frac{\ln(1/\delta)}{2(n+k)}}$$

$$\begin{aligned}
f^{\text{CH, WIS}}(D, 0, 1, k) &= \frac{1}{\beta}\sum_{i=1}^{n} w_i g_i - b\sqrt{\frac{\ln(1/\delta)}{2(n+k)}} \\
&> \frac{1}{\beta}\sum_{i=1}^{n} w_i g_i - b\sqrt{\frac{\ln(1/\delta)}{2n}} \\
&= f^{\text{CH, WIS}}(D, w_y, g_y, 0),
\end{aligned} \quad (1)$$

where (1) follows from $b\sqrt{\frac{\ln(1/\delta)}{2n}} > b\sqrt{\frac{\ln(1/\delta)}{2(n+k)}}$. Second, we show that (2) in Definition 1 holds.

$$\begin{aligned}
\frac{\partial}{\partial w_y} f^{\text{CH, WIS}}(V, w_y, g_y, k) &= -\frac{k\sum_{i=1}^{n} w_i g_i}{(kw_y + \beta)^2} - \frac{k^2 w_y g_y}{(kw_y + \beta)^2} + \frac{kg_y}{(kw_y + \beta)} \\
&= -\frac{k\sum_{i=1}^{n} w_i g_i}{(kw_y + \beta)^2} - \frac{k^2 w_y g_y}{(kw_y + \beta)^2} + \frac{kg_y(kw_y + \beta)}{(kw_y + \beta)^2} \\
&= -\frac{k\sum_{i=1}^{n} w_i g_i}{(kw_y + \beta)^2} + \frac{kg_y\beta}{(kw_y + \beta)^2} \\
&= -\frac{k\sum_{i=1}^{n} w_i g_i}{(kw_y + \beta)^2} + \frac{k\sum_{i=1}^{n} w_i g_y}{(kw_y + \beta)^2} \\
&= \frac{k}{(\beta + kw_y)^2}\sum_{i=1}^{n} w_i(g_y - g_i)
\end{aligned}$$

$$\frac{\partial}{\partial w_y} f^{\text{CH, WIS}}(V, w_y, 1, k) = \frac{k}{(\beta + kw_y)^2}\sum_{i=1}^{n} w_i(1 - g_i). \quad (2)$$

Notice that (2) is non-negative: 1) When $g_y = 1$, (2) is positive as long as there exists at least one $g_i < 1$ for $i \in \{1, \ldots, n\}$; 2) If all $g_i = 1$ in $D$, then (2) is zero. The following is the derivative of $f^{\text{CH, WIS}}(D, w_y, g_y, k)$ w.r.t. $g_y$:

$$\frac{\partial}{\partial g_y} f^{\text{CH, WIS}}(D, w_y, g_y, k) = \frac{kw_y}{(\beta + kw_y)} \quad (3)$$

$$\frac{\partial}{\partial g_y} f^{\text{CH, WIS}}(D, i^*, g_y, k) = \frac{ki^*}{(\beta + ki^*)},$$

which is also non-negative. To find any critical points, the following equations are solved simultaneously: $\partial/\partial g_y f^{\text{CH, WIS}}(D, w_y, g_y, k) = 0$ and $\partial/\partial w_y f^{\text{CH, WIS}}(D, w_y, g_y, k) = 0$. Notice that (3) is zero when $w_y = 0$. Plugging $w_y = 0$ into $\partial/\partial w_y f^{\text{CH, WIS}}(D, w_y, g_y, k) = 0$, and then solving for $g_y$, yields the $x$ coordinate of a critical point.

$$\frac{k}{(\beta + k(0))^2}\sum_{i=1}^{n} w_i(g_y - g_i) = 0$$

$$\frac{k}{\beta^2}\sum_{i=1}^{n} w_i(g_y - g_i) = 0$$

$$g_y \sum_{i=1}^{n} w_i - \sum_{i=1}^{n} w_i g_i = 0$$

$$g_y = \frac{\sum_{i=1}^{n} w_i g_i}{\beta}.$$

The following partial derivatives are computed to classify whether $(0, \sum_{i=1}^{n} w_i g_i / \beta)$ is a minimum, maximum or saddle point:

$$\frac{\partial}{\partial (w_y)^2}(D, w_y, g_y, k) = \frac{-2k^2}{(\beta + k w_y)^3} \sum_{i=1}^{n} w_i (g_y - g_i).$$

$$\frac{\partial}{\partial (g_y)^2}(D, w_y, g_y, k) = 0.$$

$$\frac{\partial}{\partial g_y w_y}(D, w_y, g_y, k) = \frac{\partial}{\partial w_y} \frac{k w_y}{(\beta + k w_y)}$$

$$= \frac{k\beta}{(\beta + k w_y)^2}.$$

Using the second partial derivative test, the critical point is substituted into the following equation:

$$\frac{\partial}{\partial (w_y)^2} \cdot \frac{\partial}{\partial (g_y)^2} - \left(\frac{\partial}{\partial g_y w_y}\right)^2 = 0 - \left(\frac{k\beta}{(\beta + k(0))^2}\right)^2$$

$$= -\left(\frac{k\beta}{\beta^2}\right)^2,$$

which is less that zero. Therefore, $(w_y = 0, g_y = \sum_{i=1}^{n} w_i g_i / \beta)$ is a saddle point. □

Next, we describe the trajectory that must be added to $D$ to execute the optimal attack.

**Definition 2 (Optimal Attack).** *An optimal attack strategy for $k > 0$ is to select*

$$\underset{H \in \mathcal{H}_{\pi_e}}{\arg \max} f^{*,\star}\big(D, w_y = w(H, \pi_e, \pi_b), g_y = g(H), k\big).$$

**Definition 3 (Optimal Trajectory).** *Given that a maximum exists, let $(a', s') \in \underset{a \in \mathcal{A}, s \in \mathcal{S}}{\arg \max} \frac{\pi_e(a,s)}{\pi_b(a,s)}$.*
*If $\frac{\pi_e(a,s)}{\pi_b(a,s)} > 1$, let $H^* = \{S_0 = s', A_0 = a', R_0 = 1, \ldots, S_{\tau-1} = s', A_{\tau-1} = a', R_{\tau-1} = 1\}$.*
*Otherwise, let $H^* = \{S_0 = s', A_0 = a', R_0 = 1\}$.*

**Theorem 1.** *For any adversarially monotonic off-policy estimator, the optimal attack strategy is to add $k$ repetitions of $H^*$ to $D$.*

*Proof.* An optimal attack strategy is equivalent to

$$\underset{H \in \mathcal{H}_{\pi_e}}{\arg \max} f^{*,\star}\big(D, w(H, \pi_e, \pi_b), g(H), k\big) = \underset{i^* \in \mathcal{I}, g^* \in [0,1]}{\arg \max} f^{*,\star}\big(D, i^*, g^*, k\big).$$

For any off-policy estimator that is adversarially monotonic, by (1) of Definition 1, there exists a $pq$ such that

$$f^{*,\star}(D, p, q, k) \geq f^{*,\star}(D, p, q, 0).$$

A return that maximizes $f^{*,\star}(D, w_y, g_y, k)$ implies that

$$\underset{g^* \in [0,1]}{\max} f^{*,\star}(D, p, g^*, k) \geq f^{*,\star}(D, p, q, k).$$

$f^{\text{CH, IS}}$ and $f^{\text{CH, WIS}}$ are non-decreasing w.r.t. the return. Therefore,

$$\underset{g^* \in [0,1]}{\arg \max} f^{*,\star}(D, p, g^*, k) = \underset{g^* \in [0,1]}{\max} g^*.$$

Setting $g^* = 1$, an importance weight that maximizes $f^{*,\star}(D, w_y, 1, k)$ implies that

$$\underset{i^* \in \mathcal{I}}{\max} f^{*,\star}(D, i^*, 1, k) \geq f^{*,\star}(D, p, 1, k).$$

$f^{\text{CH, IS}}$ and $f^{\text{CH, WIS}}$ are also non-decreasing w.r.t. the importance weight. So,

$$\underset{i^* \in \mathcal{I}}{\arg \max} f^{*,\star}(D, i^*, 1, k) = \underset{i^* \in \mathcal{I}}{\max} i^*.$$

Since the IS weight is a product of ratios over the length of a trajectory, the ratio at a single time step is maximized.

$$\max_{i^* \in \mathcal{I}} i^* = \max_{a \in \mathcal{A}, s \in \mathcal{S}} \prod_{t=0}^{\tau-1} \frac{\pi_e(A_t = a, S_t = s)}{\pi_b(A_t = a, S_t = s)}$$

$$= \begin{cases} \left( \max\limits_{a \in \mathcal{A}, s \in \mathcal{S}} \frac{\pi_e(a,s)}{\pi_b(a,s)} \right)^{\tau} & \text{if } \max\limits_{a \in \mathcal{A}, s \in \mathcal{S}} \frac{\pi_e(a,s)}{\pi_b(a,s)} > 1, \\ \max\limits_{a \in \mathcal{A}, s \in \mathcal{S}} \frac{\pi_e(a,s)}{\pi_b(a,s)} & \text{otherwise.} \end{cases}$$

To create $H^*$, if the ratio at a single time step is greater than 1, $a'$ and $s'$ is repeated for the maximum length of the trajectory, $\tau$; otherwise, $a'$ and $s'$ is repeated only for a single time step. Thus, $H^*$ represents the trajectory with the largest return and importance weight. □

Next, we show how Equations (2) and (1), that define quasi-$\alpha$-security and $\alpha$-security, respectively, apply to $L^{*,*}$. Specifically, we show that a safety test using $L^{*,*}$ as a metric is a valid safety test that first predicts the performance of $\pi_e$ using $D$, and then bounds the predicted performance with high probability. If $L^{*,*}(\pi_e, D) > J(\pi_b)$, the safety test returns True; otherwise it returns False.

**Lemma 3.** *A safety test using $L^{*,*}$ is quasi-$\alpha$-secure if* $\forall m \in \mathcal{M}, \Pr\left(L^{*,*}\big(\pi_e, m(D, k)\big) > J(\pi_b) + \alpha\right) \leq \Pr\left(L^{*,*}\big(\pi_e, D\big) > J(\pi_b)\right).$

*Proof.* For $x \in \mathbb{N}^+$, let $\mathcal{P} : \Pi \times D_n^{\pi_b} \to \mathbb{R}^x$ denote any function to predict the performance of some $\pi_e \in \Pi$, using data $D$ collected from $\pi_b$. Also, let $\mathcal{B} : \mathbb{R}^x \times [0, 1] \to \mathbb{R}$ denote any function that bounds performance with high probability, $1 - \delta$, where $\delta \in [0, 1]$. Starting with the definition of quasi-$\alpha$-security, we have that $\forall m \in \mathcal{M}$,

$$\Pr\left(\varphi\big(\pi_e, m(D, k), J(\pi_b) + \alpha\big) = \text{True}\right) \leq \Pr\left(\varphi\big(\pi_e, D, J(\pi_b)\big) = \text{True}\right)$$

$$\iff \Pr\left(\mathcal{B}\big(\mathcal{P}(\pi_e, m(D, k)), \delta\big) > J(\pi_b) + \alpha\right) \leq \Pr\left(\mathcal{B}\big(\mathcal{P}(\pi_e, D), \delta\big) > J(\pi_b)\right)$$

$$\iff \Pr\left(L^{*,*}\big(\pi_e, m(D, k)\big) > J(\pi_b) + \alpha\right) \leq \Pr\left(L^{*,*}\big(\pi_e, D\big) > J(\pi_b)\right).$$

□

**Lemma 4.** *A safety test using $L^{*,*}$ is $\alpha$-secure if* $\forall m \in \mathcal{M}, \Pr\left(L^{*,*}\big(\pi_e, m(D, k)\big) > J(\pi_b) + \alpha\right) < \delta.$

*Proof.* For $x \in \mathbb{N}^+$, let $\mathcal{P} : \Pi \times D_n^{\pi_b} \to \mathbb{R}^x$ denote any function to predict the performance of some $\pi_e \in \Pi$, using data $D$ collected from $\pi_b$. Also, let $\mathcal{B} : \mathbb{R}^x \times [0, 1] \to \mathbb{R}$ denote any function that bounds performance with high probability, $1 - \delta$, where $\delta \in [0, 1]$. Starting with the definition of $\alpha$-security, we have that $\forall m \in \mathcal{M}$,

$$\Pr\left(\varphi\big(\pi_e, m(D, k), J(\pi_b) + \alpha\big) = \text{True}\right) < \delta$$

$$\iff \Pr\left(\mathcal{B}\big(\mathcal{P}(\pi_e, m(D, k)), \delta\big) > J(\pi_b) + \alpha\right) < \delta$$

$$\iff \Pr\left(L^{*,*}\big(\pi_e, m(D, k)\big) > J(\pi_b) + \alpha\right) < \delta.$$

□

In Lemma 5, we describe a condition that must hold in order to compute a valid $\alpha$. The condition states that a valid $\alpha$ must be equal to or greater than the largest increase in the $1 - \delta$ confidence lower bound on $J(\pi_e)$ across all datasets $D \in \mathcal{D}_n^{\pi_b}$ and all attack strategies (i.e., the optimal attack).

**Lemma 5.** *A safety test using $L^{*,*}$ is quasi-$\alpha$-secure or $\alpha$-secure if* $\forall D \in \mathcal{D}_n^{\pi_b}$ *and* $\forall m \in \mathcal{M}$, $L^{*,*}\big(\pi_e, m(D, k)\big) \leq L^{*,*}(\pi_e, D) + \alpha.$

*Proof.* If $L^{*,\star}\big(\pi_e, m(D,k)\big) \leq L^{*,\star}(\pi_e, D) + \alpha$, then

$$L^{*,\star}(\pi_e, D) \geq L^{*,\star}\big(\pi_e, m(D,k)\big) - \alpha. \tag{4}$$

A safety test checks whether $L^{*,\star}(\pi_e, D) > J(\pi_b)$. When (4) holds $\forall D \in \mathcal{D}_n^{\pi_b}$ and $\forall m \in \mathcal{M}$,

$$\Pr(L^{*,\star}(\pi_e, D) > J(\pi_b)) \geq \Pr(L^{*,\star}\big(\pi_e, m(D,k)\big) - \alpha > J(\pi_b)), \tag{5}$$

and hence via algebra that

$$\Pr(L^{*,\star}\big(\pi_e, m(D,k)\big) > J(\pi_b) + \alpha) \leq \Pr(L^{*,\star}(\pi_e, D) > J(\pi_b)),$$

which, by Lemma (3), implies that a safety test using $L^{*,\star}$ is quasi-$\alpha$-secure. In the case of $\alpha$-security, by Assumption 3, we require a "safe" safety test. That is,

$$\Pr(L^{*,\star}(\pi_e, D) > J(\pi_b)) < \delta. \tag{6}$$

From the transitive property of $\geq$, we can conclude from (5) and (6) that

$$\Pr(L^{*,\star}\big(\pi_e, m(D,k)\big) - \alpha > J(\pi_b)) < \delta,$$

and hence via algebra that

$$\Pr(L^{*,\star}\big(\pi_e, m(D,k)\big) > J(\pi_b) + \alpha) < \delta,$$

which, by Lemma (4), implies that a safety test using $L^{*,\star}$ is $\alpha$-secure. $\qquad\square$

## B  Proof of Theorem 1

The result of (5) for the estimator that uses CH and IS is the following:

$$\alpha' = \max_{D \in \mathcal{D}_n^{\mathcal{H}}} f^{\text{CH, IS}}(D, i^*, 1, k) - L^{\text{CH, IS}}(\pi_e, D)$$

$$= \max_{D \in \mathcal{D}_n^{\mathcal{H}}} \frac{1}{n+k} \sum_{i=1}^{n} w_i g_i + \frac{k}{n+k}(i^*)(1) - b\sqrt{\frac{\ln(1/\delta)}{2(n+k)}} - \left( \frac{1}{n} \sum_{i=1}^{n} w_i g_i - b\sqrt{\frac{\ln(1/\delta)}{2n}} \right)$$

$$= \max_{D \in \mathcal{D}_n^{\mathcal{H}}} b\sqrt{\frac{\ln(1/\delta)}{2n}} - b\sqrt{\frac{\ln(1/\delta)}{2(n+k)}} + \frac{k}{(n+k)}\left( i^* - \frac{\sum_{i=1}^{n} w_i g_i}{n} \right).$$

Recall that $b$ represents the upper bound of all IS weighted returns. Let $b = i^*$, and $g_i = 0$ for all $i \in \{1, \ldots, n\}$.

$$\alpha' = i^* \sqrt{\frac{\ln(1/\delta)}{2n}} - i^* \sqrt{\frac{\ln(1/\delta)}{2(n+k)}} + \frac{k}{(n+k)}(i^* - 0)$$

$$= i^* \left( \sqrt{\frac{\ln(1/\delta)}{2n}} - \sqrt{\frac{\ln(1/\delta)}{2(n+k)}} + \frac{k}{(n+k)} \right).$$

The result of (5) for the estimator that uses CH and WIS is the following:

$$\alpha' = \max_{D \in \mathcal{D}_n^{\mathcal{H}}} f^{\text{CH, WIS}}(D, i^*, 1, k) - L^{\text{CH, WIS}}(\pi_e, D)$$

$$= \max_{D \in \mathcal{D}_n^{\mathcal{H}}} \frac{1}{ki^* + \sum_{i=1}^{n} w_i} \left( \sum_{i=1}^{n} w_i g_i + k(i^*)(1) \right) - b\sqrt{\frac{\ln(1/\delta)}{2(n+k)}} - \left( \frac{1}{\sum_{i=1}^{n} w_i} \sum_{i=1}^{n} w_i g_i - b\sqrt{\frac{\ln(1/\delta)}{2n}} \right)$$

$$= \max_{D \in \mathcal{D}_n^{\mathcal{H}}} b\sqrt{\frac{\ln(1/\delta)}{2n}} - b\sqrt{\frac{\ln(1/\delta)}{2(n+k)}} + \frac{ki^*}{(ki^* + \beta)}\left( 1 - \frac{\sum_{i=1}^{n} w_i g_i}{\beta} \right).$$

Let $g_i = 0$ for all $i \in \{1, \ldots, n\}$. Also, notice that $b = 1$ because importance weighted returns are in range $[0, 1]$ for WIS.

$$\alpha' = \max_{D \in \mathcal{D}_n^{\mathcal{H}}} \sqrt{\frac{\ln(1/\delta)}{2n}} - \sqrt{\frac{\ln(1/\delta)}{2(n+k)}} + \frac{ki^*}{(ki^* + \beta)}.$$

Recall that $\beta \neq 0$. So, let $w_i = 0$ for all $i \in \{1, \ldots, n-1\}$ and $w_n = i^{\min}$.

$$\alpha' = \sqrt{\frac{\ln(1/\delta)}{2n}} - \sqrt{\frac{\ln(1/\delta)}{2(n+k)}} + \frac{ki^*}{(i^{\min} + ki^*)}.$$

# C  Panacea: An Algorithm for Safe and Secure Policy Improvement

Table 1: $\alpha$–security of `Panacea`.

| Estimator | $\alpha$ |
|-----------|----------|
| CH, IS | $c\left(\sqrt{\frac{\ln(1/\delta)}{2n}} - \sqrt{\frac{\ln(1/\delta)}{2(n+k)}} + \frac{k}{(n+k)}\right)$ |
| CH, WIS | $\sqrt{\frac{\ln(1/\delta)}{2n}} - \sqrt{\frac{\ln(1/\delta)}{2(n+k)}} + \frac{kc}{(i^{\min}+kc)}$ |

---

**Algorithm 1** `Panacea`$(D, \pi_e, \alpha, k)$

---

1: Compute $c$, using $\alpha$ and $k$, given estimator
2: **for** $H \in D$ **do**
3:     **if** IS weight computed using $H$ is greater than $c$ **then**
4:         Set IS weight to $c$
5: return clipped $D$

---

## C.1  Proof of Corollary 1

Let $\alpha'$ and $k'$ denote the user-specified inputs to `Panacea`. Based on Table 1, $c^{\text{CH, IS}} = \alpha'/\left(\sqrt{\frac{\ln(1/\delta)}{2n}} - \sqrt{\frac{\ln(1/\delta)}{2(n+k)}} + \frac{k}{(n+k)}\right)$ if $k' = k$. Recall that $b$ is the upper bound on all IS weighted returns. Due to clipping, $b = c^{\text{CH, IS}}$, and let $g_i = 0$ for all $i \in \{1, \ldots, n\}$. The result of (6) for the estimator that uses CH and IS is the following:

$$\max_{D \in \mathcal{D}_n^{\mathcal{H}}} f^{\text{CH, IS}}(\texttt{Panacea}(D, c^{\text{CH, IS}}), c^{\text{CH, IS}}, 1, k) - L^{\text{CH, IS}}\left(\pi_e, \texttt{Panacea}(D, c^{\text{CH, IS}})\right)$$

$$= \max_{D \in \mathcal{D}_n^{\mathcal{H}}} \frac{1}{n+k} \sum_{i=1}^{n} w_i g_i + \frac{k}{n+k}(c^{\text{CH, IS}})(1) - b\sqrt{\frac{\ln(1/\delta)}{2(n+k)}} - \left(\frac{1}{n}\sum_{i=1}^{n} w_i g_i - b\sqrt{\frac{\ln(1/\delta)}{2n}}\right)$$

$$= \max_{D \in \mathcal{D}_n^{\mathcal{H}}} b\sqrt{\frac{\ln(1/\delta)}{2n}} - b\sqrt{\frac{\ln(1/\delta)}{2(n+k)}} + \frac{k}{(n+k)}\left(c^{\text{CH, IS}} - \frac{\sum_{i=1}^{n} w_i g_i}{n}\right)$$

$$= c^{\text{CH, IS}}\sqrt{\frac{\ln(1/\delta)}{2n}} - c^{\text{CH, IS}}\sqrt{\frac{\ln(1/\delta)}{2(n+k)}} + \frac{k}{(n+k)}(c^{\text{CH, IS}} - 0)$$

$$= c^{\text{CH, IS}}\left(\sqrt{\frac{\ln(1/\delta)}{2n}} - \sqrt{\frac{\ln(1/\delta)}{2(n+k)}} + \frac{k}{(n+k)}\right)$$

$$= \frac{\alpha'}{\sqrt{\frac{\ln(1/\delta)}{2n}} - \sqrt{\frac{\ln(1/\delta)}{2(n+k)}} + \frac{k}{(n+k)}} \cdot \left(\sqrt{\frac{\ln(1/\delta)}{2n}} - \sqrt{\frac{\ln(1/\delta)}{2(n+k)}} + \frac{k}{(n+k)}\right)$$

$$= \alpha'.$$

For WIS, recall that no matter how the clipping weight is set, $b \leq 1$ because importance weighted returns are in range $[0, 1]$, and $\beta \neq 0$. So, let $w_i = 0$ for all $i \in \{1, \ldots, n-1\}$ and $w_n = i^{\min}$. Also, let $g_i = 0$ for all $i \in \{1, \ldots, n\}$. Based on Table 1, $c^{\text{CH, WIS}} = i^{\min}\left(\alpha' - \sqrt{\frac{\ln(1/\delta)}{2n}} + \sqrt{\frac{\ln(1/\delta)}{2(n+k)}}\right)/k\left(1 - \alpha' + \sqrt{\frac{\ln(1/\delta)}{2n}} - \sqrt{\frac{\ln(1/\delta)}{2(n+k)}}\right)$ if $k' = k$. The result of (6) for the estimator that uses

CH and WIS is the following:

$$\max_{D \in \mathcal{D}_n^{\mathcal{H}}} f^{\text{CH, WIS}}(\text{Panacea}(D, c^{\text{CH, WIS}}), c^{\text{CH, WIS}}, 1, k) - L^{\text{CH, WIS}}(\pi_e, \text{Panacea}(D, c^{\text{CH, WIS}}))$$

$$= \max_{D \in \mathcal{D}_n^{\mathcal{H}}} \frac{1}{kc^{\text{CH, WIS}} + \sum_{i=1}^{n} w_i} \left( \sum_{i=1}^{n} w_i g_i + k(c^{\text{CH, WIS}})(1) \right) - b\sqrt{\frac{\ln(1/\delta)}{2(n+k)}} - \left( \frac{1}{\sum_{i=1}^{n} w_i} \sum_{i=1}^{n} w_i g_i - b\sqrt{\frac{\ln(1/\delta)}{2n}} \right)$$

$$= \max_{D \in \mathcal{D}_n^{\mathcal{H}}} b\sqrt{\frac{\ln(1/\delta)}{2n}} - b\sqrt{\frac{\ln(1/\delta)}{2(n+k)}} + \frac{kc^{\text{CH, WIS}}}{(kc^{\text{CH, WIS}} + \beta)} \left( 1 - \frac{\sum_{i=1}^{n} w_i g_i}{\beta} \right)$$

$$\leq \sqrt{\frac{\ln(1/\delta)}{2n}} - \sqrt{\frac{\ln(1/\delta)}{2(n+k)}} + \frac{kc^{\text{CH, WIS}}}{(kc^{\text{CH, WIS}} + i^{\min})}$$

$$= \left( \sqrt{\frac{\ln(1/\delta)}{2n}} - \sqrt{\frac{\ln(1/\delta)}{2(n+k)}} \right) + \frac{\left( \alpha' - \sqrt{\frac{\ln(1/\delta)}{2n}} + \sqrt{\frac{\ln(1/\delta)}{2(n+k)}} \right)}{\left( \alpha' - \sqrt{\frac{\ln(1/\delta)}{2n}} + \sqrt{\frac{\ln(1/\delta)}{2(n+k)}} \right) + \left( 1 - \alpha' + \sqrt{\frac{\ln(1/\delta)}{2n}} - \sqrt{\frac{\ln(1/\delta)}{2(n+k)}} \right)}$$

$$= \sqrt{\frac{\ln(1/\delta)}{2n}} - \sqrt{\frac{\ln(1/\delta)}{2(n+k)}} + \alpha' - \sqrt{\frac{\ln(1/\delta)}{2n}} + \sqrt{\frac{\ln(1/\delta)}{2(n+k)}}$$

$$= \alpha'.$$

## C.2 Proof of Corollary 2

Let $\alpha$ and $k'$ denote the user-specified inputs to $\text{Panacea}$. If $k' = k$, i.e., the user inputs the correct number of trajectories added by the attacker, the result of (6) for the estimator that uses CH and IS is the following:

$$\alpha = \max_{D \in \mathcal{D}_n^{\mathcal{H}}} f^{\text{CH, IS}}(\text{Panacea}(D, c), c, 1, k) - L^{\text{CH, IS}}(\pi_e, \text{Panacea}(D, c))$$

$$\alpha = c \left( \sqrt{\frac{\ln(1/\delta)}{2n}} - \sqrt{\frac{\ln(1/\delta)}{2(n+k)}} + \frac{k}{(n+k)} \right)$$

$$c = \frac{\alpha}{\left( \sqrt{\frac{\ln(1/\delta)}{2n}} - \sqrt{\frac{\ln(1/\delta)}{2(n+k)}} + \frac{k}{(n+k)} \right)}.$$

If $k' = k$, the result of (6) for the estimator that uses CH and WIS is the following:

$$\max_{D \in \mathcal{D}_n^{\mathcal{H}}} f^{\text{CH, WIS}}(\text{Panacea}(D, c), c, 1, k) - L^{\text{CH, WIS}}(\pi_e, \text{Panacea}(D, c)) \leq \sqrt{\frac{\ln(1/\delta)}{2n}} - \sqrt{\frac{\ln(1/\delta)}{2(n+k)}} + \frac{kc}{(kc + i^{\min})}.$$
(7)

Setting the right-hand side of (7) to $\alpha$, and solving for $c$ equals:

$$\alpha = \sqrt{\frac{\ln(1/\delta)}{2n}} - \sqrt{\frac{\ln(1/\delta)}{2(n+k)}} + \frac{kc}{(i^{\min} + kc)}$$

$$\frac{kc}{(i^{\min} + kc)} = \alpha - \sqrt{\frac{\ln(1/\delta)}{2n}} + \sqrt{\frac{\ln(1/\delta)}{2(n+k)}}$$

$$kc - kc\alpha + kc\sqrt{\frac{\ln(1/\delta)}{2n}} - kc\sqrt{\frac{\ln(1/\delta)}{2(n+k)}} = i^{\min}\alpha - i^{\min}\sqrt{\frac{\ln(1/\delta)}{2n}} + i^{\min}\sqrt{\frac{\ln(1/\delta)}{2(n+k)}}$$

$$kc \left( 1 - \alpha + \sqrt{\frac{\ln(1/\delta)}{2n}} - \sqrt{\frac{\ln(1/\delta)}{2(n+k)}} \right) = i^{\min}\alpha - i^{\min}\sqrt{\frac{\ln(1/\delta)}{2n}} + i^{\min}\sqrt{\frac{\ln(1/\delta)}{2(n+k)}}$$

$$c = \frac{i^{\min} \left( \alpha - \sqrt{\frac{\ln(1/\delta)}{2n}} + \sqrt{\frac{\ln(1/\delta)}{2(n+k)}} \right)}{k \left( 1 - \alpha + \sqrt{\frac{\ln(1/\delta)}{2n}} - \sqrt{\frac{\ln(1/\delta)}{2(n+k)}} \right)}.$$