[Reviews · NeurIPS 2020]

Review 1

Summary and Contributions: The paper studies the robustness of Seldonian RL algorithms to adversarial attacks, devising an algorithm that is both safe and secure. A specific concept of security is introduced, addressing an attacker that can trick the Seldonian algorithm into accepting an unsafe policy by injecting adversarial trajectories in the off-policy evaluation phase. The proposed solution is evaluated on a medical problem.

Strengths: The problem of combining safety and security is very interesting and, I think, also novel and important. Safe RL is often designed for tasks where unpredictable behavior could cause serious problems, including harm to people. These are also the scenarios in which an adversarial attack could do more damage.

Weaknesses: The scope is a bit narrow. The work focus on a specific class of safe RL algorithms, that are Seldonian algorithms. It identifies a specific vulnerability and a specific kind of attack. It is still interesting, but a broader discussion on the combination of safety and security would be valuable.

Correctness: I did not check the proofs in the appendix, but the methodology seems sound.

Clarity: Yes, the presentation is good and I found no typos or unclear sentences.

Relation to Prior Work: Yes, the Related Work section is short but it does its job.

Reproducibility: Yes

Additional Feedback: Update: thanks for the answer. I strongly encourage you to further discuss the scope of your work in the final version, as you outlined in the rebuttal. I also would like to stress once again that mentioning Hoeffding's inequality without further details is confusing. ---- 1. Line 34: it is not clear at this point what "corrupted data" are 2. You assumed finite states and actions for simplicity. How do your results generalize to continuous MDPs? 3. Line 53: do you mean "almost surely"? 4. Line 101: IS yields unbounded weighted returns, WIS is biased. In both cases, I see problems in applying Hoeffding's inequality 5. Footnote 2: I would swap \pi_b and \pi_e 6. What happens if the attacker knows we are using Panacea? 7. Line 243: I do not see the purpose of mentioning per-decision importance sampling in a setting where it is equivalent to standard IS


Review 2

Summary and Contributions: The authors observe that offline policy evaluation algorithms are not robust with respect to adversarial attacks, i.e. dataset corrupted with counterfeited samples. They concentrate on the worst case scenario that consists in the samples that are constructed such that the policy evaluation overestimates the performance, which would lead the Seldonian RL algorithm to accept a new policy, although it is actually worse than the baseline policy. The authors analytically issue an upper bound for the security of classic importance sampling algorithms (namely IS and WIS) and propose to improve them by performing an additional clipping to the weighting of the samples. Indeed, the worst case scenario happens when the importance sampling weighting gets very large. The theoretical results are supported with experiments.

Strengths: S1: The research subject is well grounded in requirements for having RL actually applicable to real world problems. S2: The authors have made an effort to both mathematically formalize their setting and to extensively explain it with text. As a result, the submission is clear and didactic.

Weaknesses: W1: The study seems to focus too much on algorithms that are based on safety tests. Algorithms such as that of [Petrik2016] or [Laroche2019] also offer Seldonian or quasi-Seldonian guarantees without relying on a safety test. I understand that the analysis is not compatible, but maybe that would be worth it to include studies on how easy it is to trick those algorithms too. More generally (even for IS algorithms), it was a bit odd to me that the study does not consider attacks on the way pi_e is chosen. W2: It's unclear to me whether the trajectory must still have been performed in the real environment, or it can be completely be made up (but then its value has to be within the range [0,1]). Also, with model based methods (for both environment and policy models), it might be possible to single out the few trajectories that are inconsistent with the other trajectories. W3: The discounted return of any trajectory is assumed to belong to [0,1]. So it means that J belongs also to [0,1]. Wouldn't it mean that the use of any alpha>=1 would be pointless? The bounds in Table 1 seem to be almost always larger than 1, and the experiments report some results with alpha>=1. Is there something I misunderstood?

Correctness: The paper is correct to the best of my knowledge/understanding.

Clarity: I really enjoyed the effort on clarity. The few points that remained unclear to me are detailed in the weaknesses section.

Relation to Prior Work: The references are diverse and relevant. The only criticism I have with this regard is the one of W1: I would include a discussion on the security of algorithms that are do not use safety tests. My guess would be that they are not robust at all and a counter example should be easy to construct. [Petrik2016] Petrik, M., Ghavamzadeh, M., & Chow, Y. (2016). Safe policy improvement by minimizing robust baseline regret. In Advances in Neural Information Processing Systems (pp. 2298-2306). [Laroche2019] Laroche, R., Trichelair, P., & Des Combes, R. T. (2019, May). Safe policy improvement with baseline bootstrapping. In International Conference on Machine Learning (pp. 3652-3661).

Reproducibility: Yes

Additional Feedback: Despite its flaws, I feel that this submission is a good contribution to the field of Safe/Robust RL and I recommend its acceptance. Minor remarks: M1: Lines 52-53: Assuming a finite horizon seems to be a strong impractical assumption. Couldn't it be relaxed? M2: Footnote 2: Shouldn't it be pi_e absolutely continuous with pi_b? M3: Line 149: Shouldn't the attack function be a function of the 7 items that are enumerated in the previous paragraph? M4: Definition 3: Tightness is barely used. It seems unnecessary to formalize it. M5: Corollary 1: There is no Table in the appendix, and there is not Appendix D. (but lines 269-271 were clear about what is in this table) __________________________________________________ __________________________________________________ __________________________________________________ Post-rebuttal comments: The rebuttal cleared up some minor misunderstanding or uncertainty I had. I keep my score at 7, and raise my confidence to 5.


Review 3

Summary and Contributions: The paper discusses certain Seldonian RL algorithms, in which, while attempting to build better policies, new candidate policies are compared to the previous best-known policy using a "safety test"; if the new policy tests as likely better, it is kept, while otherwise, the new policy is discarded. The paper considers an attacker attempting to fool this safety test by adding fraudulent data to the trajectories used for the test to make the new policy artificially seem better, leading the algorithm to generate suboptimal policies. The paper defines the concept of such a test being resistant to these attacks by being "(quasi-)alpha-secure," and prove under what values of hyperparameters certain such tests are alpha-secure. It then produces an algorithm which uses this construction to be provably secure against these adversarial attacks. It shows that in a couple RL domains, the new algorithm does not demonstrate the same brittleness to altered data as the original Seldonian approach.

Strengths: - The paper is carefully done, and notation is admirably consistent and well-defined in a paper that has a lot of work to do to get the reader up to speed to understand the contributions. This paper was written by talented scientists. - Safe and secure RL are soon to be areas of extreme importance. - The empirical evaluations do show well that the new algorithm is much more robust to adversarial attacks on the training set.

Weaknesses: - Significance seems limited. Seldonian RL is niche, and these results are only proven on a subset of Seldonian algorithms. To make sure my unfamiliarity with it wasn't unusual, I checked the citation counts of the works this one stands on the shoulders of, and they were low. While this is a deeply flawed measure of quality or future significance, when combined with my unfamiliarity, it does suggest impact will likely be limited. I worry it represents a deeper and deeper exploration of a domain only a very few are interested in - It is not clear to me there are conclusions or insights I can apply to other RL algorithms. I didn't learn anything that didn't only apply very specifically to the (uncommon) algorithms being discussed. - The attack being defended against is not presented as realistic, but as a worst-case stand-in for real-world data issues. However, it's difficult to know how much worse the worst-case is compared to realistic issues.Without the attack being realistic, or the real-world problems it represents being discussed further, it is hard to know if the algorithm is worth the effort.

Correctness: I believe so, and the careful writing has earned my trust on the appendices I didn't carefully read.

Clarity: The writing is very good. It was not an easy read, however, because, though I'm very familiar with a lot of work in RL, I had a lot to learn to understand the contributions. I believe this will be a common experience.

Relation to Prior Work: I am not adequately familiar with Seldonian RL work to answer this well.

Reproducibility: Yes

Additional Feedback: POST AUTHOR FEEDBACK: Thank you for the response. I still suspect this paper is perhaps too niche for NeurIPS, but as I said, it is well written. I look forward to the promised expanded context in your introduction! In anticipation of that, I have raised my score to a 6.


Review 4

Summary and Contributions: This paper presents an analysis of a class of safe RL algorithms, dubbed Seldonian, in the context of assumption-violating data errors. They introduce a measure called $\alpha$-security to assess the robustness of security tests in Seldonion RL systems and show how even minor data corruption -- specifically the introduction of corruption examples adversarially -- can invalidate guarantees. They propose an alternative approach called Panacea and evaluate it on threeexamples.

Strengths: - Well motivated: a clear and important problem is addressed. - The paper is well written and structured - The panacea algorithm is straightforward to implement and consequently the method has the potential to be immediately useful in many domains.

Weaknesses: The Panacea approach requires as input an upper bound on the number of corrupted samples. It seems that how realistic this is, and how tight the bound in, can vary wildly across domains. It is unclear whether the practical limitations as alluded to in the conclusion -- namely that the clipping weight increases with $\alpha$ but decreases with increasing $k$ -- prevents the approach from being useful in practical domains.

Correctness: I did not check the proofs in the appendix, but the main body appears sound.

Clarity: The paper is clear and very well written. It introduces concepts in a concise manner. I appreciated the unusual but appropriate placement of the related work section. For readers less familiar with the robust RL framework some more context would be welcome, for instance in explaining up front the rationale behind the behavioral policy to begin with and its connection to safety.

Relation to Prior Work: The work is well situated within the context of other approaches to robust reinforcement learning. It is less clear how it relates to other approaches to verifying safety properties, explored more by the formal methods and verification community, but which has explored more into RL-type domains in recent years.

Reproducibility: Yes

Additional Feedback: # Typos Line 112: Some work models ...

[Author Response · NeurIPS 2020]

Thank you for your insightful feedback, corrections, and additional references that we will incorporate into our paper.
Below, we address several key points.

**All Reviewers (particularly R3):** "scope is a bit narrow" & "significance seems limited" - Based on consistent feedback
from the reviewers, we see that our presentation currently limits the perceived relevance and general importance of
our work. In our introduction, we will emphasize that our work is directly applicable to any scenario that requires
computing confidence intervals around importance sampling (IS) estimates. More broadly, we will also discuss that the
community is interested in our definition of safety [38] and its limitations [39], and IS [4 (in paper), 40, 43, 44]. Lastly,
we will mention that the $\alpha$-security formalization also pertains to high-confidence methods that do not use IS [31 (in
paper), 41, 42].

**R1:** "generalize to continuous MDPs" - Our work generalizes to continuous MDPs, but care must be taken to select $\pi_e$
such that IS weights are bounded. For example, the diabetes treatment simulation discussed in the paper has continuous
states and actions. We will make sure to discuss this extension in the paper.

"problems in applying Hoeffding's inequality" - In order to use Hoeffding's inequality, we assume that the IS weights
are bounded. Although WIS is biased, it works very well in practice. Consequently, we introduced the notion of
quasi-$\alpha$-security in Definition 2 to specifically allow for the analysis of WIS.

"if the attacker knows we are using Panacea" - The optimal attack does not change (lines 261–262), and therefore,
Panacea limits the damage incurred by the attacker. We will move this discussion outside of the proof block.

**R2**: "include studies on how easy it is to trick those algorithms too" - We are definitely interested in pursuing follow-up
directions to ensure security for model-based approaches, which we predict would be quite different and a significant
contribution on its own. In a future work section, we will include a discussion of similarities and additional challenges
that arise in that setting.

"the way $\pi_e$ is chosen" - We completely agree on the importance of how $\pi_e$ is chosen, even though the violation of
safety comes primarily from the safety test. Our current definition of security assumes that $\pi_e$ has lower performance
than $\pi_b$, but does not specify how often this occurs. Attacking the data used to select $\pi_e$ can increase this frequency.
Notice that attacking the data used to select $\pi_e$ alone would not cause the safety property to be violated. We will add a
discussion on this topic.

"whether the trajectory must still have been performed in the real environment" & "single out the few trajectories" -
Because the transition and reward functions are not known, one can not distinguish real and fake trajectories. Rare
events are critical to account for, and may look like fake trajectories. Perhaps impossible trajectories can be identified
using domain-specific knowledge, but that must be analyzed on a per-domain basis. We will mention this in the paper.

"use of any $\alpha \geq 1$ would be pointless" - We see that we did not provide sufficient discussion of Table 1, saying that the
behavior you note is what we aim to show! The middle column is usually $\geq 1$, indicating that standard methods can be
completely broken (make pessimal policies appear optimal) easily, as you described. However, the right column shows
values of $c$ that make Panacea $\alpha$-secure for *any* $\alpha \in [0, 1]$. E.g., plugging in $\alpha = 0.05$ gives Panacea a meaningful
security guarantee. Note that if $c \leq 0$, Panacea is not useful, but that the values of $c$ are positive and grow quickly as $n$
grows relative to $k$.

**R3:** "a worst-case stand-in" - When the stakes are high – for example, in the application of RL to sepsis treatment in
the intensive care unit, wherein training data is generated from hand-written doctors' notes – we do not want to assume
that the data contains only minor errors (such as patient height), but also major ones (such as wrong drug name).

**R4:** "an upper bound on the number of corrupted samples" - We will add a discussion of the many issues faced by
practitioners, including estimating the number of corrupt trajectories (perhaps based on known error rates in the data
processing pipeline of NLP and computer vision models) and selecting $\pi_e$. Panacea is only one piece of the puzzle, but
provides guarantees that are informative to practitioners.

**References:** [38] Ghavamzadeh, Mohammad et al. Safe policy improvement by minimizing robust baseline regret.
NeurIPS 2016; [39] Guo, Zhaohan et al. Using options and covariance testing for long horizon off-policy policy
evaluation. NeurIPS 2017; [40] Jiang, Nan et al. Doubly robust off-policy value evaluation for reinforcement learning.
ICML 2016; [41] Kuzborskij, Ilja et al. Confident Off-Policy Evaluation and Selection through Self-Normalized
Importance Weighting. arXiv preprint arXiv:2006.10460 2020; [42] Laroche, Romain et al. Safe policy improvement
with baseline bootstrapping. ICML 2019; [43] Liu, Qiang et al. Breaking the curse of horizon: Infinite-horizon
off-policy estimation. NeurIPS 2018; [44] Mandel, Travis et al. Offline policy evaluation across representations with
applications to educational games. AAMAS 2014.


[Meta-Review · NeurIPS 2020]

All the reviewers support acceptance for the contributions, notably improvements to the robustness of RL algorithms to adversarial attacks, and a clear exposition on how these methods can be applied to real world problems. I also recommend acceptance. Please consider revising the paper to address the concerns raised in the reviews and rebuttal, in particular to better explain the scope of the work. Separately, it may be useful to extend the broader impact statement to inform a casual reader that a mathematical safety guarantee on an algorithm is not a replacement for domain specific safety requirements (for example, the diabetes treatment would still need oversight for medical safety).